# USING FORWARDS-BACKWARDS MODELS TO APPROXIMATE MDP HOMOMORPHISMS

## ABSTRACT

Animals are able to rapidly infer, from limited experience, when sets of state-action pairs have equivalent reward and transition dynamics. On the other hand, modern reinforcement learning systems must painstakingly learn through trial and error that sets of state-action pairs are value equivalent—requiring an often prohibitively large amount of samples from their environment. MDP homomorphisms have been proposed that reduce the observed MDP of an environment to an abstract MDP, which can enable more sample efficient policy learning. Consequently, impressive improvements in sample efficiency have been achieved when a suitable MDP homomorphism can be constructed a priori—usually by exploiting a practitioner's knowledge of environment symmetries. We propose a novel approach to constructing a homomorphism in discrete action spaces, which uses a learnt partial model of environment dynamics to infer which state-action pairs lead to the same state—reducing the size of the state-action space by a factor equal to the cardinality of the action space. On MDP homomorphism benchmarks, we demonstrate improved sample efficiency over previous attempts to learn MDP homomorphisms, while achieving comparable sample efficiency to approaches that rely on prior knowledge of environment symmetries. In MinAtar, we report an almost 4x improvement in score in the low sample limit, compared to a value based baseline when averaging over all games and optimizers.

## 1 INTRODUCTION

Reinforcement learning (RL) agents outperform humans on previously impregnable benchmarks such as Go (Silver et al., 2016) and Starcraft (Vinyals et al., 2019). However, the computational expense of RL hinders its deployment in real world applications. State of the art deep RL agents can demand hundreds of millions of samples (or even hundreds of billions) to learn a policy—either within an environment model (Hafner et al., 2020) or by direct interaction (Badia et al., 2020; Fan et al., 2023). While sample efficient approaches continue to improve, there is still a significant gap between the best performing agents and those which are sample efficient (Schwarzer et al., 2023).

In contrast, humans can abstract away details about state-action pairs that do not effect their values. Consider the Atari game Asterix, where an agent collects treasure while avoiding enemies in a 2D navigation task—a human gamer understands that collecting treasure when approaching from the left has the same value as collecting the same treasure but approaching from the right. In this work, we identify this abstraction as a Markov decision process (MDP) homomorphism (Ravindran and Barto, 2001; van der Pol et al., 2020b). MDP homomorphisms collapse equivalent state-actions in an observed MDP onto a single abstract state-action pair in an abstract MDP (van der Pol et al., 2020b).

Given a mapping from experienced MDP to an abstract MDP, policies can be learned efficiently in the smaller abstract space and then mapped back to the experienced MDP when interacting with the environment (Ravindran and Barto, 2001). Previous works hard code homomorphisms into a policy (van der Pol et al., 2020b) but developing approaches that learn homomorphic mappings from experience is an active area of research (Rezaei-Shoshtari et al., 2022; Keurti et al., 2023).

We developed *equivalent effect abstraction*, a method for constructing MDP homomorphisms from experience via a dynamics model—leveraging the fact that state-action pairs leading to the same next state frequently have equivalent values. Consider again the Asterix example—moving to a given state has the same value whether your previous action was right, left, up or down. Thus, equipped with a

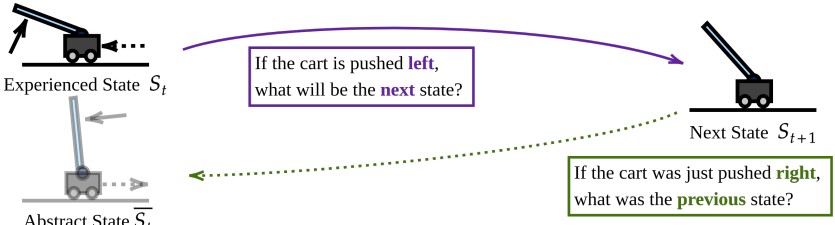

Figure 1: We find equivalent states by moving forward and then backward through a model of the environment. Unlike previous approaches (van der Pol et al., 2020b), we are *not* exploiting the horizontal symmetry of the environment. In contrast, we take advantage of the fact that state-action pairs that lead to the same state usually have equivalent values.

dynamics model, we show that if the value of approaching a state from the right is known, one can also deduce the value of approaching the same state from the left, below, or above. Consequently, equivalent effect abstraction extrapolates value judgements between equivalent state-action pairs, reducing the amount of experience required to learn a policy. We make the following contributions.

1. We develop a novel approach for constructing MDP homomorphisms (equivalent effect abstraction) that requires no prior knowledge of symmetries.

2. In the tabular setting, we show equivalent effect abstraction improves the planning efficiency of model-based RL and the sample efficiency of model-free RL.

3. In the deep RL setting, equivalent effect abstraction can be learned from experience for Cartpole and Predator-Prey environments (which are MDP homomorphisms benchmarks from van der Pol et al. (2020b)).

4. Across the MinAtar suite (Young and Tian, 2019), we demonstrate an almost 4x improvement in the low sample limit averaged over different optimizers and games.

We are not the first to exploit the equivalent values of state action pairs that lead to the same next state. Tesauro et al. (1995) use the concept of afterstates (states immediately before an opponents response to an action (Antonoglou et al., 2021)) in their backgammon program. This is similar to our approach and has been scaled within MuZero to other board games (Antonoglou et al., 2021), but in general afterstates have not found wider applicability. Misra et al. (2020)'s kinematic inseparability is a related but distinct abstraction that uses contrastive learning to classify whether two states will lead to the same next state for the *same* action. In contrast, our approach looks for state-action pairs (with *different actions*) that lead to the same next state, which is a broader form of abstraction. Another difference is that our equivalent effect abstraction does not require states to be discrete, which makes it readily applicable to common RL environments where it is rare to reach the same state twice. Finally Anand et al. (2015) defines a state-action equivalence that is more comprehensive than the equivalence defined by Ravindran and Barto (2001), which is satisfied by various equivalent state-action pairs like those found in Ravindran and Barto (2001); van der Pol et al. (2020b); Misra et al. (2020) as well as the state-action pairs satisfying the equivalent effect abstraction we present. However, they assume a tree of discrete states can be searched over for equivalences, which is challenging in deep RL.

## 2 BACKGROUND: MDP HOMOMORPHISMS

Using the definition from Silver (2015), an MDP $\mathcal{M}$ can be described by a tuple $\langle \mathcal{S}, \mathcal{A}, \mathcal{P}, \mathcal{R}, \gamma \rangle$ where $\mathcal{S}$ is the set of all states, $\mathcal{A}$ is the set of all actions, $\mathcal{R} = \mathbb{E}[R_{t+1}|S_t = s, A_t = a]$ is the reward function that determines the scalar reward received at each state, $\mathcal{P} = \mathbb{P}[S_{t+1} = s'|S_t = s, A_t = a]$ is the transition function of the environment describing the probability of moving from state to another for a given action and $\gamma \in [0, 1]$ is the discount factor describing how much an agent should favour immediate rewards over those in future states. An agent interacts with an environment through its policy $\pi(a|s) = \mathbb{P}[A_t = a|S_t = s]$ (Silver, 2015) which maps the current state to a given action. To solve an MDP, an RL agent must develop a policy that maximises the expected return $G$, which is equal to the sum of discounted future rewards $G = \mathbb{E}_\pi[\sum_{t=0}^{T} \gamma^t R_{t+1}]$ (where $t$ is the current timestep and $T$ is the number of timesteps in a learning episode).

Ravindran and Barto (2001) introduced the concept of a homomorphism which, using the notation and definitions from van der Pol et al. (2020b), is a homomorphism $h = \langle \{\sigma_a | a \in \mathcal{A}\}, \{\alpha_s | s \in \mathcal{S}\}\rangle$ from an agent's experienced MDP $\mathcal{M}$ to an abstract MDP $\bar{\mathcal{M}} = \langle \bar{\mathcal{S}}, \bar{\mathcal{A}}, \bar{\mathcal{P}}, \bar{\mathcal{R}}, \gamma\rangle$—satisfying the following, $\forall s, s' \in \mathcal{S}$ and $a, a' \in \mathcal{A}$:

$$\bar{\mathcal{P}}(\sigma_{a'}(s') | \sigma_a(s), \alpha_s(a)) = \sum_{s'' \in \sigma_{a'}^{-1}(s')} \mathcal{P}(s'' | s, a) \tag{1}$$

$$\bar{\mathcal{R}}(\sigma_a(s), \alpha_s(a)) = \mathcal{R}(s, a) \tag{2}$$

Where $\sigma$ is a mapping from experienced state-action pairs to abstract states $\sigma : \mathcal{S}_a \rightarrow \bar{\mathcal{S}}_{\alpha_s(a)}$ dependent on action $a$, while $\alpha_s : \mathcal{A} \rightarrow \bar{\mathcal{A}}_{\sigma_a(s)}$ is a state dependent mapping between experienced and abstract actions and $a'$ the action taken when in the next state $s'$ (Ravindran and Barto, 2001; van der Pol et al., 2020b). Note, we have made one update to the definition from van der Pol et al. (2020b), which is that $\sigma_a$ now depends on state and actions instead of just states. In the deep learning setting, our states are vector representations while are our actions are one-hot encodings of discrete actions. Importantly, in contrast to other homomorphism approaches (e.g. (van der Pol et al., 2020a; Rezaei-Shoshtari et al., 2022)), our state mapping predicts abstract states in the form that they are emitted from the environment, rather than compressing them to a lower dimensionality.

Equations (1) and (2) show that a homomorphic map maintains the transition dynamics and reward functions of the experienced MDP within the constructed abstract MDP. As proved in Ravindran and Barto (2001) and leveraged by van der Pol et al. (2020b), if the Q-values in the abstract MDP are known, $\bar{Q}^*(\sigma_a(s), \alpha_s(a))$, then equipped with a homomorphism $h$, the true Q-values in the experienced MDP can be "lifted" (Ravindran and Barto, 2001) from their abstract counterparts.

$$\bar{Q}^*(\sigma_a(s), \alpha_s(a)) = Q^*(s, a) \quad \forall s \in \mathcal{S}, a \in \mathcal{A} \tag{3}$$

Where $Q$ is the value of state $s$ and action $a$ and $*$ signifies the value function is optimal (Ravindran and Barto, 2001), where again we have used the notation from (van der Pol et al., 2020b). The size of abstract state-action space $\bar{\mathcal{S}} \times \bar{\mathcal{A}}$ is often smaller than the size of the experienced state-action space $\mathcal{S} \times \mathcal{A}$. Consequently, if an agent is equipped with a homomorphic map, a policy can be learned efficiently in the abstract state-action space and then "lifted" to the experienced state-action space (Ravindran and Barto, 2001). In general, obtaining a homomorphic map is difficult. Next, we present a novel method for inferring a homomorphic map from experience.

## 3 EQUIVALENT EFFECT ABSTRACTION

Equivalent effect abstraction is based on a simple observation that is guaranteed to be true in the majority of RL problems if $\mathcal{R}$ and $\mathcal{P}$ are deterministic. Firstly, state-action pairs that lead to the same next state have equivalent reward functions in the majority of RL testbeds (e.g. DeepMind control (Tassa et al., 2018), classic control (Brockman et al., 2016) and Atari (Bellemare et al., 2013)). While common MDP formulations (like the one used in section 2) define reward functions as depending on the current state and previous action, in many environments it is possible to drop the dependence on the previous action. Nevertheless, we note that we assume no dependence on the previous action for the reward function, so our approach is not suitable for optimal control settings where different actions may be defined to have a different cost (Liberzon, 2011, p. 145). Next, we develop a method to learn the mapping $\sigma_a$ between state-action pairs that lead to the same state.

Consider an agent that takes an action $a_t$ in a state $s_t$ and ends up in state $s_{t+1}$, once in state $s_{t+1}$, how can we infer what other state-action pairs could also lead to state $s_{t+1}$? We could make this inference if we had both a forwards *and* backwards model of the environment. To train our forwards model $\mathcal{F}$ and backwards model $\mathcal{B}$ (where color represents correspondence to prediction directions in figure 1), we collect experience tuples $\langle s_t, a_t, s_{t+1}\rangle$ from the environment and then optimise the parameters of our models $\theta$ and $\phi$ with a MSE loss function.

$$\arg\min_{\theta} \|\mathcal{F}_\theta(s_t, a_t) - s_{t+1}\|_2^2 \tag{4}$$

$$\arg\min_{\phi} \|\mathcal{B}_\phi(s_{t+1}, a_t) - s_t\|_2^2 \tag{5}$$

---

**Algorithm 1** Computing Equivalent Effect Abstraction Q-Values

---

**Inputs:** Forward Model $\mathcal{F}_\theta$, Backward Model $\mathcal{B}_\phi$, List of $N$ environment actions $A_N$, hypothetical action $a_{hyp}$, environment state $s$, value function $Q$
**Output:** List of predicted Q-values $q_N$
 1: Initialize list of Q-values $q_N$
 2: **for** $i = 0$ to $N$ **do**
 3:     **if** $A_N[i] = a_{hyp}$ **then**
 4:         $q_N[i] \leftarrow Q(s, a_{hyp})$
 5:     **else**
 6:         $s' \leftarrow \mathcal{F}_\theta(s, A[i])$               ▷ predict one step forward for $i^{th}$ action and input state $s$
 7:         $\bar{s} \leftarrow \mathcal{B}_\phi(s', a_{hyp})$                  ▷ predict hypothetical state $\bar{s}$ for $a_{hyp}$
 8:         $q_N[i] \leftarrow Q(\bar{s}, a_{hyp})$
 9:     **end if**
10: **end for**

---

Where in the deep RL setting forwards model $\mathcal{F}_\theta$ and backwards model $\mathcal{B}_\phi$ are parametrised models that take concatenated state-action vectors as input and then output state vectors. In the tabular setting, these models are lookup tables. Throughout the paper, if models are learned, they are trained with the simple mean-squared error loss shown above, with experience tuples sampled from a DQN replay buffer. We compute Q-values using algorithm 1, otherwise our approach is equivalent to standard Q-learning (Watkins and Dayan, 1992) (in the tabular setting) or DQN (Silver et al., 2016) (in the deep RL setting).

With these models we are ready to reduce the size of our MDP. First, we set the abstraction action $\bar{a}$ to always be our *hypothetical* action $a_{hyp}$. Next for a given state-action pair $(\mathbf{s}, a)$ the equivalent (hypothetical) state-action pair can be computed by moving forwards through our model and then backwards with the hypothetical action $a_{hyp}$—meaning equivalent effect abstraction's MDP homomorphism is defined as follows.

$$\sigma(s_t, a_t) = \mathcal{B}_\phi(\mathcal{F}_\theta(s_t, a_t), a_{hyp}) \qquad\qquad \alpha_s(a) = a_{hyp} \qquad\qquad (6)$$

### 3.1 Equivalent effect abstraction delivers an approximate MDP homomorphism

Now we demonstrate how equivalent effect abstraction is MDP homomorphism by applying equations (1) and (2) to one step transition tuples $(s, a, r, s')$. As we assume deterministic dynamics, the transition homomorphism (equation (1)) simplifies to the following.

$$\bar{\mathcal{P}}(\sigma_{a'}(s')|\sigma_a(s), \alpha_s(a)) = \mathcal{P}(s'|s, a) \qquad\qquad (7)$$

Where our state mapping $\sigma$ also depends on the current action when applying its homomorphic mapping. Now we substitute our definition of the homomorphic state and action mappings for equivalent effect abstraction into the left hand side of equation (7) .

$$\bar{\mathcal{P}}(\bar{s}'|\bar{s}, a_{hyp}) = \mathcal{P}(s'|s, a) \qquad\qquad (8)$$

Recall that the forwards and backwards model learn to predict an abstract state, that for a given hypothetical action deterministically transitions to the next state $s'$—meaning equation (1) holds, as both $(\bar{s}, a_{hyp})$ and $(s, a)$ transition to $s'$ with certainty (and by definition, $\bar{s}' = s'$).

We also make the assumption that reward functions are defined for state-action pairs by the state that they transition into (i.e. if two state action pairs transition into a goal state, it does not matter from what direction they arrived at the goal state). This means that equation (2) is rewritten as $\bar{\mathcal{R}}(\sigma_{a'}(s')) = \mathcal{R}(s')$. As $\sigma_a$ is equal to the identity for next states, equation (2) also holds for equivalent effect abstraction. It is important to stress that as our mapping satisfies equation (3), it also assumes the MDP homomorphism properties developed by Ravindran and Barto (2001), meaning the policy we learn with our hypothetical state action pairs can be "lifted" van der Pol et al. (2020b) and used as a policy in a given experienced environment.

## 3.2 Hypothetical Actions

In theory, one could calculate multiple equivalent state-action pairs for a given state-action pair by querying the backwards model with multiple different actions. However, our aim is to simplify the state-action space that our policy learns in, so instead we select one action to be our "hypothetical" action and all other state-action pairs are mapped into its reference frame. In Figure 1, the hypothetical action is moving right—meaning our Q-network only has to learn about moving right and we can map all right values to their equivalent left values with our forwards and backwards models. The hypothetical action is a hyperparameter. In some environments such as Cartpole, the hypothetical action can be selected randomly. However, in other environments backwards state predictions are not always possible for a given state-action pair—meaning hypothetical actions need to be selected more carefully.

By finding an equivalent hypothetical state-action pair for each experienced state-action pair we reduce the number of state-action pairs that an off-policy algorithm must learn to estimate—reducing the size of the space of values from $\mathcal{S} \times \mathcal{A}$ to $\mathcal{S} \times 1 + N$, where $N \in \mathcal{S} \times \mathcal{A}$ are the states where the hypothetical action is not computable because of edge cases that we discuss next.

## 3.3 When Equivalent Effect Abstraction Fails

We have assumed that for every state-action pair, at least one equivalent hypothetical state-action can be computed. This is often true but it is not guaranteed. For example, near borders in a grid world there is no way to travel left to a border state that has a border on its right. We only found this to significantly effect results in the tabular setting, which we mitigate by reverting to vanilla Q-learning table when necessary. This backup Q-learning value table is learned in parallel. Furthermore, in stochastic environments it may be difficult to predict equivalent states. In section 6, we suggest recurrent state space models could be used to sample from a distribution of predicted transitions in future work (Hafner et al., 2019). Lastly, when forward dynamics are deterministic the previous state could still be unpredictable as a unique previous state for a given previous action may not exist (Yu et al., 2021). Again the model based RL provides some ideas for addressing this (Yu et al., 2021).

## 4 Experiments

We test our approach on a range of tasks, from tabular RL to using a convolutional DQN (Silver et al., 2016). Where possible, we try to have overlap in our experiments with the MDP homomorphism literature—using Cartpole to overlap with the well known van der Pol et al. (2020b) and van der Pol et al. (2020a) as well as Predator Prey to overlap with van der Pol et al. (2020b). Hyperparameter details can be found in the Appendix. Shaded regions indicate the standard error.

### 4.1 Tabular Maze

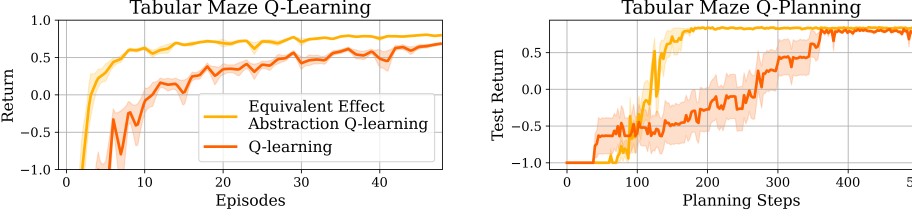

Figure 2: (a), (b) In a gridworld, removing redundancies in the Q-table improves sample efficiency. For the vanilla baselines, an agent must learn the value of each state-action pair for all actions in the action space. With equivalent effect abstraction, we reduce the number of Q-values that need to be learned. This yields improvements in both model-free Q-learning and model-based approaches Q-planning. 50 seeds are used for Q-learning while 10 seeds are used for Q-planning.

We begin with the maze environment from Sutton and Barto (2018, p. 165). The maze consists of $6 \times 9$ cells. The agent starts on the far west of the maze and must navigate east and find its way around walls in the middle of the environment. After passing the walls, the agent must then travel further east to reach a corridor, before moving north to reach a goal location. In this particular experiment, we assume a model of the environment is known beforehand—which we move through forwards and then backwards to create a homomorphic map.

We use an open-source Q-learning (Watkins and Dayan, 1992) implementation as a baseline[1] (along with the maze code from the same authors). We then adapt the Q-learning implementation to make use of a homomorphic map as shown in Algorithm 1. We show in figure 2(a) that our approach compares favorably to vanilla Q-learning, converging much faster to the optimal policy. In the gridworld environment, we reduce the size of the state action space and hence the number of Q-values (when going left) from 216 to 67. In this case, the number of states in the reduced action space is not reduced by the cardinality of the action space exactly. This is because of the 13 states where it is not possible to reach them by traveling left because there is a border to the right.

To test planning efficiency, we performed a further experiment on the gridworld with model-based Q-planning (Sutton and Barto, 2018, p. 161). Our Q-planning method is equivalent to the Q-learning implementation but altered to learn by randomly sampling model transitions. At each Q-planning update we pause training and evaluate performance in a 100 step episode. Planning performance is shown in figure 2(b). Similarly to the Q-learning results, leveraging knowledge of equivalent actions is able to significantly improve learning efficiency.

## 4.2 CARTPOLE

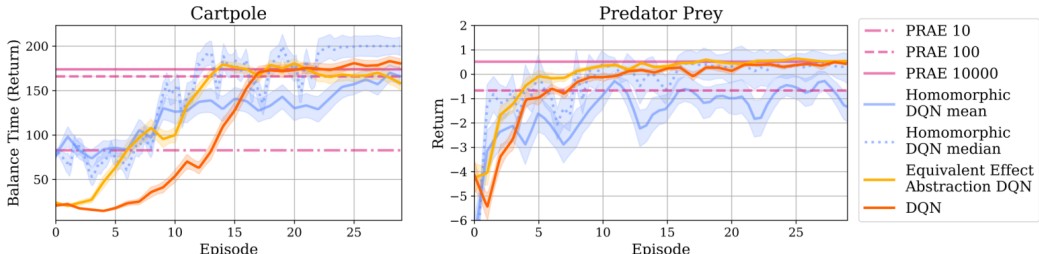

Figure 3: (**a**) Cartpole: we improve upon the DQN baseline significantly (the model is learned in the initial 3 episodes of experience—a different 3 episodes per repeat—*that is shown on the x-axis of the plot*). For vanilla DQN and equivalent effect abstraction we use 50 seeds, 5 seeds are used for each PRAE result and 30 seeds are used for MDP homomorphic networks. For clarity, PRAE 10 refers to using PRAE with 10 episodes of data to construct an environment model, which it then plans in. (**b**) Predator Prey: we learn a mapping to improve sample efficiency over vanilla DQN. Unlike in Cartpole *the experience used to train the homomorphic map is not included in the plot* for equivalent effect abstraction DQN. For equivalent effect abstraction we reuse one set of pre-trained backwards and forwards models for each RL run.

Next, we apply our approach to the Cartpole benchmark (Brockman et al., 2016). In Cartpole, an agent controls a cart with a pendulum attached to it. The goal of the task is to learn to balance the pendulum upright by moving the cart horizontally left or right. The states are a four-dimensional vector (position, velocity, angle, angular velocity) and Cartpole has a discrete action space (move the cart left or right). To learn a model, we use 3 episodes of learning experience at the beginning of training to train the homomorphic map (two simple linear models, one for each action optimised with Adam (Kingma and Ba, 2015)) to generate equivalent state-action pairs for the opposite action to the one experienced. These initial training steps *are* included for equivalent effect abstraction in Figure 3(a). We integrate the learned homomorphism into a DQN implementation (Silver et al., 2016)[2], with Q-values only being learned for an arbitrarily chosen hypothetical action.

---

[1] github.com/thehawkgriffith/dyna-maze
[2] https://gist.github.com/Pocuston/13f1a7786648e1e2ff95bfad02a51521

We compare our approach to three baselines: vanilla DQN, MDP homomorphic Networks (van der Pol et al., 2020b) and PRAE (van der Pol et al., 2020a). For all experiments we use the baselines' open source implementations and the libraries they build upon (Stooke and Abbeel, 2019). MDP homomorphic networks use specially constructed weights that are equivariant to environment symmetries (requiring prior knowledge of symmetries). PRAE trains a contrastive model to learn a mapping to a latent "plannable" MDP, that satisfies the definitions of an MDP homomorphism. PRAE then performs planning on the learned abstract MDP. For Cartpole, we learned a mapping for PRAE with datasets of 10, 100, 10000 episodes of random experience. We plot the average convergence performance of PRAE's planning algorithm. In the Cartpole task, we did not need to avoid any hypothetical state action pairs that are not computable—meaning a reduction of the size of the state action space from $\mathcal{S} \times \mathcal{A}$ to $\mathcal{S} \times 1$.

As shown in figure 3(a), in the low sample regime we improve upon both PRAE and vanilla DQN—with our approach converging at around episode 12 while vanilla Q-learning takes around 16 episodes to converge—note that this improvement includes the number of episodes required to learn our mapping. We found that in many cases MDP homomorphic networks were able to achieve good performance but worst case runs brought down mean performance significantly. To demonstrate this, we plot the mean and median for MDP homomorphic networks (only the mean is plotted for other methods). The median MDP homomorphic network is also able to similarly improve upon the vanilla DQN baseline, by leveraging a practitioner's prior knowledge of environment symmetries. MDP homomorphic networks are not necessarily a competing approach to equivalent effect abstraction and future work could conceivably combine these two approaches for even greater sample efficiency.

### 4.3 STOCHASTIC PREDATOR PREY

Following the seminal work of van der Pol et al. (2020b), we also benchmark our method on the Predator Prey environment, where an agent must chase a stochastically moving prey in a 2D world (van der Pol et al., 2020a). The observations are a $7 \times 7 \times 3$ tensor encoding agent position and prey position. The objective of the agent is to catch the prey in fewest steps possible. Rewards are set to -0.1 unless the predator catches the prey, in which case the episode ends and a reward of 1 is provided.

For our method, we train action-dependent forwards and backwards models on a dataset of transitions created by taking random actions in the environment for $10^4$ environment steps (equivalent to around 170 episodes of random experience). Note that the environment is stochastic, so learning a perfect environment model is impossible. We compare to the baselines introduced in section 4.2. For PRAE, we benchmark with both 10000 and 100 episodes of experience data and perform planning to convergence. Similar to the Cartpole results, PRAE can be effective, but only when a relatively large amount of model training experience is available.

In Predator Prey, a policy is learned with around 5 episodes with a vanilla DQN. As a result, in this particular environment it is not practical to use equivalent effect abstraction as by the time a model can be learned the policy has already converged. However, what figure 4(b) does show is that if a model can be obtained before online policy learning, then equivalent effect abstraction can deliver an improvement over vanilla DQN—even though the environment is stochastic. This is consistent with previous literature on MDP homomorphisms (van der Pol et al., 2020b), which designed the Predator Prey environment to test whether their MDP homomorphism was robust to stochasticity.

### 4.4 MINATAR

Our previous results were on relatively low-dimensional environments. In this section we test whether EEA can be deployed on the MinAtar suite (Young and Tian, 2019)—a more accesible version of the Atari benchmark (Bellemare et al., 2013) where observations are image-like tensors. MinAtar has a relatively high dimensionality when compared to the environments used in previous sections. Furthermore, MinAtar is interesting because it violates many of the assumptions made in the theoretical construction of equivalent effect abstraction. Firstly, MinAtar has sticky actions (Machado et al., 2018)—meaning predicting the next state correctly is impossible even if your environment model is perfect. Secondly, various MinAtar games have states where predicting the equivalent hypothetical state is impossible, such as in border states in Breakout.

Table 1: Normalized scores for MinAtar 250K

| game | actions removed | DQN | EEA | DYNA | homomorphic DQN |
|------|-----------------|-----|-----|------|------------------|
| Asterix | 4 | 0.093 ± 0.015 | **0.3 ± 0.056** | 0.05 ± 0.0058 | 0.076 ± 0.0099 |
| Breakout | 2 | 0.71 ± 0.052 | 0.77 ± 0.036 | 0.72 ± 0.046 | 0.65 ± 0.077 |
| Freeway | 2 | 0.7 ± 0.02 | 0.72 ± 0.023 | 0.037 ± 0.025 | 0.018 ± 0.012 |
| Seaquest | 4 | 0.031 ± 0.0074 | **0.11 ± 0.02** | 0.039 ± 0.0056 | 0.046 ± 0.0043 |
| space invaders | 2 | 0.69 ± 0.028 | 0.66 ± 0.032 | 0.79 ± 0.079 | 0.71 ± 0.061 |

In our experiments, we do not make any corrections for stochasticity or impossible to compute hypothetical border states. The only inductive bias we inject is the hard-coding of the hypothetical actions. For Asterix and Breakout, we select "no-op" as our hypothetical action, Freeway uses "up" as its hypothetical action while in Seaquest and Space Invaders we use two hypothetical actions "fire" and "no-op" (because there is equivalent moving/no-op action for fire). We train a U-Net style architecture (Ronneberger et al., 2015) for our forward and backwards models that is integrated into our equivalent effect abstraction agent (see appendix A.3.3 for details on architecture). We train these models continuously in parallel with our Q-network, making model updates with samples from the replay buffer after each Q-learning step.

We include three different baselines, the vanilla DQN implementation that we build upon from Young and Tian (2019), a homomorphic DQN (van der Pol et al., 2020b) and a model-based Dyna-DQN agent. We implement a Dyna agent that augments the DQN baseline with additional experience from a learned model for a direct comparison with equivalent effect abstraction. While other model-based baselines exist (e.g. Hafner et al. (2019)), they are on-policy and actor-critic based making them an unfair comparison to our off-policy value-based DQN. We also note that the state of the art in discrete control is a off-policy value based agent, which outperforms all model-based approaches in the low-sample regime (Schwarzer et al., 2023). As a result, if we can improve the sample efficiency of off-policy value-based methods, then there is potential to advance upon the state-of-the-art for discrete control. Our Dyna agent makes forward state, reward and episode termination predictions—it then augments its value network experience with these rollouts.

Full hyperparameters for each algorithm are described in appendix A.2.4. We find that picking the best optimizer for EEA and DQN depends on the environment and the number of frames available (we discuss how to interpret this in the next paragraph). For all other methods we report performance with RMSprop. Furthermore, we report normalised scores, where we divide the score by the performance of a DQN in the high-sample limit ($2.5 \times 10^6$ frames). Inspired by the Atari 100k benchmark (Kaiser et al. (2019)), we benchmark performance at 250k frames.

When comparing the performance of EEA and DQN using the RMSprop optimizer, we find that EEA generally gives a performance boost in the low sample limit (Freeway, Seaquest and Asterix) or matches the DQN's performance (Breakout and Space Invaders). On the other hand, when using the Adam optimizer this improvement is not as clear—such as in Freeway, where DQN and EEA's performance is equal in the low sample limit and better than the performance of the RMSprop agents. However, using the Adam optimizer can also cause huge performance drops for both EEA and DQN on other games. For example, while in Seaquest using the Adam optimizer gives an initial boost in performance, the peak performance is roughly 80% worse than using an RMSprop optimzer for DQN (see figure 5 for a full breakdown of results). This picture is relatively nuanced—to

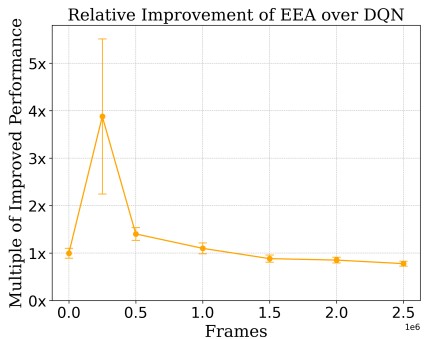

Figure 4: Equivalent effect abstraction delivers an almost 4x performance boost in the low sample limit.

provide a more clear conclusion we compute how much better the EEA performs compared to the DQN for each optimizer (where we compute the ratio of EEA RMSprop performance to DQN

Table 2: Peak performance across all frames

| game | actions removed | DQN | EEA | DYNA | homomorphic DQN |
|---|---|---|---|---|---|
| Asterix | 4 | **1 ± 0.059** | 0.85 ± 0.083 | 0.67 ± 0.047 | 0.69 ± 0.1 |
| Breakout | 2 | 1.2 ± 0.098 | 1 ± 0.11 | 1 ± 0.17 | 1.1 ± 0.11 |
| Freeway | 2 | **1 ± 0.019** | 0.96 ± 0.015 | 0.79 ± 0.022 | 0.97 ± 0.0064 |
| Seaquest | 4 | 1.1 ± 0.17 | 0.91 ± 0.18 | 1.1 ± 0.26 | 0.8 ± 0.13 |
| Space Invaders | 2 | 1.2 ± 0.09 | 1.1 ± 0.093 | 0.79 ± 0.079 | **1.4 ± 0.099** |

RMSprop performance and the ratio of EEA Adam performance to DQN Adam performance). We then average these ratios over all games and plot them against training frame budgets in figure 4. This figure shows, on average, **an almost 4x improvement over DQN in the low sample limit**.

In table 1, we compare EEA to our baselines (where EEA and DQN performances are quoted with Adam, the best optimizer in the low sample limit for EEA and DQN). At 250k frames, equivalent effect abstraction outperforms all baselines on 2/5 games while matching the other best algorithm's performance on 2/5 games and only being significantly worse than than the best algorithm (Dyna) on one game. While all games have 6 actions, the number of *effective* actions may be smaller. For example, in Freeway the agent cannot move left or right nor can you fire a projectile—meaning the number of actions collapsed by equivalent effect abstraction is only 2 ("no-op" and "down"). Accordingly, our method outperforms other methods when the number of effective actions removed is large (e.g. Asterix, where an agent moves in all cardinal directions and these actions can be collapsed down to the hypothetical action "no-op").

**Tradeoffs of equivalent effect abstraction in the large sample limit.** Equivalent effect abstraction maintains respectable performance compared to other methods as the number of samples increases but does not improve overall performance when many samples are available. Results in table 2 use the RMSprop optimizer for DQN and EEA as it performs best given a large sample limit. We attribute the weak performance of EEA in the large sample limit to the fact that many MinAtar environments do not obey the assumptions we made in the theoretical development of our method. As a result, there is a trade-off to be made when deciding whether to use equivalent effect abstraction. On one hand, when samples are scarce, equivalent effect abstraction can improve performance. On the other hand, stochasticity, imperfect models and uncomputable hypothetical states mean that baselines overtake equivalent effect abstraction as the number of samples increases.

## 5 RELATED WORK

Work on MDP homomorphisms was initialised by Ravindran and Barto (2001) whoe developed a framework for abstraction under symmetry with theoretical guarantees. Givan et al. (2003) proposed model minimisation using bisimulation (Larsen and Skou, 1991), where states could be mapped to an abstract state if their transition dynamics and reward sequences were indistinguishable. These early homomorphisms required prior knowledge from a practitioner.

Biza et al. (2021) use bisimulation to compute structured hidden Markov model priors to infer a reduced state space. Bisimulation can be relaxed to a similarity metric (Ferns et al., 2011; Ferns and Precup, 2014)—these metrics have been used to condition embeddings with contrastive losses to generate invariant representations. Zhang et al. (2020) augment an embedding encoder with an additional contrastive loss term that results in encoder distances forming an approximate bisimulation metric—allowing agents to perform control tasks in the presence of distractors. Rezaei-Shoshtari et al. (2022) formalise continuous homomorphisms—building a homomorphism through a bisimulation metric. Keurti et al. (2023) use an autoencoder architecture that learns representations that respect group actions, but do not apply their approach to MDPs. Zhu et al. (2022) use a contrastive loss and define state-action pairs with the same actions as equivalent in an abstract space. Agarwal et al. (2021) outline a policy similarity metric that is used in conjunction with a contrastive loss—pulling together embeddings that result in similar trajectories. In general, contrastive learning achieves impressive results but can be sample hungry (van der Pol et al., 2020a).

Homomorphisms in MDPs have also been used in tasks with symmetries. PRAE (van der Pol et al., 2020a) trained a contrastive world model that satisfies the transition dynamics definition equation (1)) by design. van der Pol et al. (2020b) learn network weights that are equivariant to environment symmetries but their algorithm requires a practitioner to hardcode symmetric groups beforehand. Similar approaches have been adopted for environments with continuous symmetries (Wang et al., 2022). Biza and Platt (2019) approach the problem of finding MDP homomorphisms using online partition iteration—predicting which partition a state should fall into given an action, and refining the partitions through splitting. Asadi et al. (2019) and Lyu et al. (2023) explore equivalences where state-actions pairs lead to separate states but with the same probabilities.

Perhaps the most relevant related approach is the use of afterstates (Sutton and Barto, 2018, p. 136). Afterstates slightly shift an MDP out of phase with conventional state transitions, creating environment states that are in-between the initial effect of a policy's action and the reaction of the environment to said action. Applications of the afterstates framework have generally been constrained to board games (Tesauro et al., 1995) and focus on dealing with the stochasticity of an opponent rather than improved sample efficiency (Antonoglou et al., 2021). Also closely related, Misra et al. (2020) elegantly deploy a similar concept—using contrastive learning to learn an abstraction that groups together state-action pairs that will pass through or have passed through same state with the same previous action. However, this approach is not necessarily scalable to common RL environments such as Atari and classic control tasks where experiencing the exact same state twice is unlikely.

More broadly, model-based RL has enabled sample-efficient superhuman performance in Atari (Hafner et al., 2020), but the number of planning steps required to learn a policy is still large. Backwards models have been proposed to improve sample efficiency of world model representation learning (Yu et al., 2021), which would be interesting if integrated with equivalent effect abstraction.

## 6 LIMITATIONS AND FUTURE WORK

Stochastic dynamics can make environment transitions multimodal, meaning point predictions of future or previous states are not adequate. In theory, this can be dealt with using stochastic models (Hafner et al., 2019) of forwards and backwards dynamics, that could be sampled when abstracting state-action pairs. Even worse, dynamics can be completely unpredictable (Kendall and Gal, 2017; Mavor-Parker et al., 2021), in which case uncertainty predictions could be used to signal that equivalent effect abstraction should temporarily revert to vanilla Q-learning. Further, backwards transitions may be unpredictable because an equivalent state does not exist—see section 1(Yu et al., 2021). Uncertainty predictions could also be used to avoid out-of-distribution mappings (Kendall and Gal, 2017). There is a silver lining for our approach, which is that only one step transition models are required, avoiding the challenge of learning multi-step models (Saxena et al., 2021).

Equivalent effect abstraction is formulated within the framework of value based RL using discrete actions. While many of the recent breakthroughs in deep RL are value based (e.g. (Silver et al., 2017; Badia et al., 2020)), actor-critic approaches (Schulman et al., 2017; Mnih et al., 2016) are often the natural choice for control tasks. Embedding equivalent effect abstraction into actor-critic architectures is a fruitful avenue for future research. It is also conceivable to formulate equivalent effect abstraction within a continuous action space by simply discretising the action space, which has allowed discrete action space methods to obtain state of the art performance on RL tasks (Banino et al., 2021). Additionally, equivalent effect abstraction could also be integrated into existing homomorphic MDP methods that rely on symmetries to further reduce the size of the abstract state-action space (van der Pol et al., 2020b). In practice, this would mean using a homomorphic Q-network (van der Pol et al., 2020b) but modified to only have one hypothetical action.

## 7 CONCLUSION

Equivalent effect abstraction is a simple method that reduces the size of a state-action space. It is easy to implement and it requires no prior knowledge of environment symmetries. We show that equivalent effect abstraction can improve the sample efficiency of policy learning in tabular environments, control tasks with continuous state spaces and stochastic deep RL environments. An exciting next

step would be to integrate equivalent effect abstraction into model-based RL (Hafner et al., 2020; Yu et al., 2021) to improve planning efficiency.

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

# A   APPENDIX

## A.1   IMPACT STATEMENT

We do not believe that EEA introduces any novel dangers or concerns in terms of broader impact. Like all reinforcement learning algorithms, our agents likely do not generalise very well and could behave unexpectedly when deployed in real world environments. We suggest extreme caution when using any RL algorithm for consequential applications.

## A.2   HYPERPARAMETER SEARCH

Below we show the hyperparameters swept through for homomorphic MDP, DQN and Equivalent Effect Abstraction agents, broken down by environment. The PRAE architectures were generally kept the same as the hyperparameters provided in (van der Pol et al., 2020a), with the exception of the learning rate which we evaluate at $0.0001, 0.001$ and $0.1$

### A.2.1   SUTTON AND BARTO TABULAR GRIDWORLD

We use the hyperparameters specified in (Sutton and Barto, 2018, p. 165): namely, Learning Rate $= 0.1$, $\gamma = 0.95$ and $\epsilon = 0.1$

### A.2.2   CARTPOLE

Table 3: Hyperparameters swept through for the Cartpole environment. Learning rate decay refers to decaying the learning rate by a factor of ten at after a specified number of episodes have elapsed.

| Hyperparameter | Values |
| --- | --- |
| Learning Rate | $0.00001, 0.0001, 0.001, 0.01$ |
| $\epsilon$ decay schedule | No decay, exponential $\tau = \frac{-1}{200}$ |
| $\gamma$ | $0.8, 0.99$ |
| Activation | ReLU, tanh |
| Learning Rate decay | No decay, 5, 10, 15, 20 |

**Homomorphic MDP** best hyperparameters: Learning Rate = 0.001, $\epsilon$ decay schedule = No decay, $\gamma = 0.8$, activation = tanh, Learning Rate decay = No decay

**Equivalent Effect Abstraction** best hyperparameters: Learning Rate = 0.001, $\epsilon$ decay schedule = No decay, $\gamma = 0.8$, activation = tanh, Learning Rate decay = 10

**Vanilla DQN** best hyperparameters: Learning Rate = 0.001, $\epsilon$ decay schedule = No decay, $\gamma = 0.8$, activation = tanh, Learning Rate decay = 15

### A.2.3   STOCHASTIC PREDATOR PREY

Table 4: Hyperparameters swept through for the Predator Prey environment.

| Hyperparameter | Values |
| --- | --- |
| Learning Rate | $0.0001, 0.001, 0.01$ |
| $\gamma$ | $0.8, 0.99$ |

**Homomorphic MDP** best hyperparameters: Learning Rate = 0.001, $\gamma = 0.99$

**Equivalent Effect Abstraction** best hyperparameters: Learning Rate = 0.01, $\gamma = 0.8$

**Vanilla DQN** best hyperparameters: Learning Rate = 0.001, $\gamma = 0.99$

### A.2.4 MINATAR

All hyperparameters were kept at the tuned DQN values provided in Young and Tian (2019) except for the optimizer for DQN and EEA, which we experimented with using Adam and RMSprop. For the homomorphic DQN, the symmetry groups breakdown as follows:

Table 5: Symmetry groups used for the homomorphic DQN (van der Pol et al., 2020b) on MinAtar.

| Game | Symmetry Group |
|---|---|
| Asterix | P4 |
| Breakout | R2 |
| Freeway | R2 |
| Seaquest | P4 |
| Space Invaders | R2 |

The groups for the homomorphic DQN were chosen by looking at gameplay footage and estimating what symmetry groups would be most useful.

For the model learning (in both EEA and Dyna), we found a learning rate of 0.0001 with an Adam optimizer was optimal. For Dyna, we tried 1 and 3 step rollouts as well as different replay ratios and found a replay ratio of 1 and rollout length of 1 to have the highest mean overall scores.

The returns provided by training runs for MinAtar be noisy—to make the performance of the algorithms at different numbers of frames clear we pause each training run and perform 30 evaluations to get the performances we quote in the tables and figures in the main text.

## A.3 MODEL ARCHITECTURES

### A.3.1 CARTPOLE

Listing 1: Homomorphic MDP Network (van der Pol et al., 2020b)

```
BasisLinear*(repr_in=4, channels_in=1, repr_out=2, channels_out=64)
ReLU() / tanh()
BasisLinear(repr_in=2, channels_in=64, repr_out=2, channels_out=64)
ReLU() / tanh()
BasisLinear(repr_in=2, channels_in=64, repr_out=2, channels_out=1)
```

*BasisLinear refers to the symmeterised layers used in (van der Pol et al., 2020b) to create homomorphic networks. This network is identical to the Cartpole network presented in that paper, but with only one output head that outputs state-action values.

Listing 2: Value Network architecture for DQN and Equivalent Effect Abstraction

```
Linear(input_size=4, output_size=1024)
tanh()
Linear(input_size=1024, output_size=1024)
tanh()
Linear(input_size=8, output_size=1024)
tanh()
Linear(input_size=1024, output_size=2)
```

Listing 3: Transition Model Architecture for Equivalent Effect Abstraction

```
Linear(input_size=2, output_size=2)
```

Listing 4: PRAE Architectures (van der Pol et al., 2020a)

```
# state encoder
Linear(input_size=4 ,output_size=64)
ReLU()
```

```
Linear(input_size=64, output_size=32)
ReLU()
Linear(input_size=32, output_size=50)
#action encoder
Linear(input_size=54 ,output_size=100)
ReLU()
Linear(input_size=100, output_size=2)
# reward prediction network
Linear(input_size=50 ,output_size=64)
ReLU()
Linear(input_size=64, output_size=1)
```

### A.3.2 PREDATOR PREY

Listing 5: Homomorphic MDP Network (van der Pol et al., 2020b)

```
BasisConv2d(repr_in=1, channels_in=1, repr_out=4, channels_out=4,
filter_size=(7,7), stride=2, padding=0)
ReLU()
BasisConv2d(repr_in=4, channels_in=4, repr_out=4, channels_out=8,
filter_size=(5,5), stride=1, padding=0)
ReLU()
GlobalMaxPool()
BasisLinear(repr_in=4, channels_in=8, repr_out=4, channels_out=128)
ReLU()
BasisLinear(repr_in=4, channels_in=8, repr_out=4, channels_out=128)
ReLU()
BasisLinear(repr_in=4, channels_in=128, repr_out=5, channels_out=1)
```

This is again the same network used in van der Pol et al. (2020b), albeit with a different output head.

Listing 6: Value Network Architecture for DQN and Equivalent Effect Abstraction

```
Linear(input_size=441, output_size=1024)
ReLU()
Linear(input_size=1024, output_size=8)
ReLU()
Linear(input_size=8, output_size=1024)
ReLU()
Linear(input_size=1024, output_size=5)
```

Listing 7: Transition Model Architecture for Equivalent Effect Abstraction

```
Linear(input_size=882, output_size=512)
ReLU()
Linear(input_size=512, output_size=8)
ReLU()
Linear(input_size=8, output_size=512)
ReLU()
Linear(input_size=512, output_size=441)
```

Listing 8: PRAE Architectures (van der Pol et al., 2020a)

```
# state encoder
Linear(input_size=441, output_size=64)
ReLU()
Linear(input_size=64, output_size=32)
ReLU()
Linear(input_size=32, output_size=50)
#action encoder
Linear(input_size=54, output_size=100)
ReLU()
Linear(input_size=100, output_size=2)
```

```
# reward prediction network
Linear(input_size=50, output_size=64)
ReLU()
Linear(input_size=64, output_size=1)
```

### A.3.3  MINATAR

Listing 9: Homomorphic MDP Network (van der Pol et al., 2020b)

```
BasisConv2d(repr_in=1, channels_in=4, repr_out=2, channels_out=16,
filter_size=(3,3), stride=1, padding=0)
ReLU()
Linear(input_size=1408, output_size=256)
ReLU()
Linear(input_size=256, output_size=6)
```

Listing 10: Value Network Architecture for DQN and Equivalent Effect Abstraction

```
Conv2d(channels_in=4, channels_out=16, filter_size=(3,3), stride=1, padding=0)
ReLU()
Linear(input_size=1024, output_size=128)
ReLU()
Linear(input_size=128, output_size=6)
```

Listing 11: MinAtar Environment Model U-net

```
Linear(input_size=10 * 10 * num_channels + num_actions, output_size=800)
ReLU()
Linear(input_size=800, output_size=10 * 10 * num_channels)
ReLU()
Linear(input_size=10 * 10 * num_channels, output_size=10 * 10 * num_channels)
```

Where there is a skip connection from the input to output layer.

Listing 12: MinAtar Reward Model

```
Linear(10 * 10 * self.num_channels + num_actions, 500)
Linear(input=500, output=10 * 10 * self.num_channels)
Linear(input=10 * 10 * self.num_channels, output=2)
```

# B   FULL MINATAR RESULTS

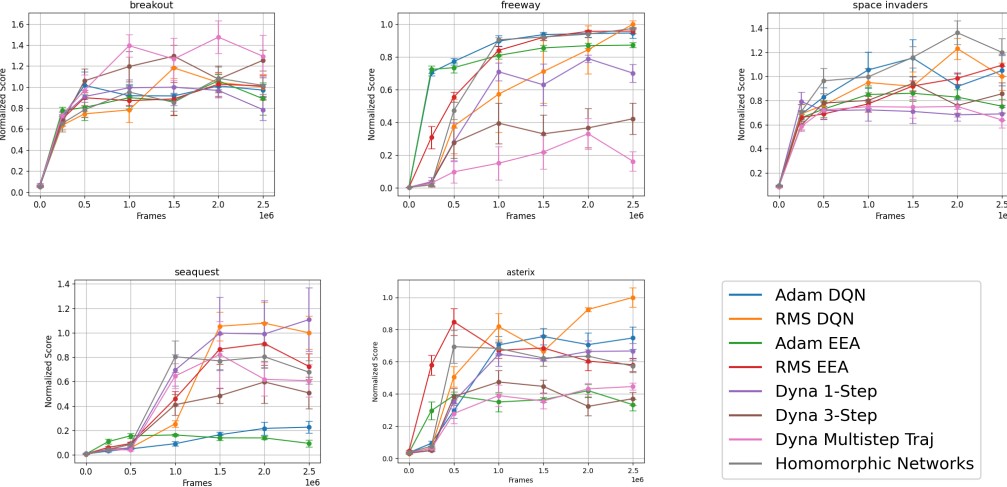

Figure 5: Full results for all different configurations of equivalent effect abstraction and the other baselines. Equivalent effect abstraction with the Adam optimizer generally performs best in the low sample limit.

