# OpenReview forum: "Using Forwards-Backwards Models to Approximate MDP Homomorphisms"
_ICLR.cc/2024/Conference — Submitted to ICLR 2024_

### Official Review · Reviewer_YpK3 · 2023-10-24

**Soundness:** 1 poor
**Presentation:** 1 poor
**Contribution:** 2 fair
**Rating:** 3
**Confidence:** 3

**Summary:**

This work proposes a forwards-backwards model to learn MDP homomorphism in discrete action and deterministic settings. Experiments are conducted on tabular gridworlds, cartpole, Predator Prey, Asterix (in MinAtar), to show its effectiveness over Homomorphic DQN and DQN.

**Strengths:**

The method is original and looks simple. The experiments on toy tasks are effective.

**Weaknesses:**

The assumptions are a bit limiting – deterministic dynamics and state-dependent rewards.

One major weakness is that I don’t know the connection between the forward and backward model (Eq 4 and 5), and MDP homomorphism (Eq 1 and 2). Why not directly learn Eq 1 and 2 (like my question below)? In addition, I don’t understand Eq 6. What is sigma(s,a), not sigma(s)? No definition. Also the hypothetical action part.

The writing quality is fair.
1. Notation writing can be improved. Be complete on $\mathcal R(s,a)$ and $\mathcal P(s’\mid s,a)$ in the background. $G$ should be in expectation.
2. Please use \citet when you cite work used as a noun.
3. Some sentences are hard to grasp. E.g. “which” and “homomorphism” are used twice at the beginning of page 3. Do not start a sentence with a math symbol like “\forall”.
4. Define the symbols like \sigma and \alpha_s explicitly with inputs and outputs (you can check how van der Pol et al., 2020b define them).
5. Typos like: Ln 7 in algorithm 1 the hat s, “Figure 3(a)” should be Figure 2(a).

The experiments are only conducted on 4-5 tasks, which are not enough for evaluating an algorithm (as the experiments are the bulk of this work). Asterix results are not in favor of the algorithm, and more MinAtar environments are expected to run for a comprehensive comparison.

**Questions:**

1. Any references on the first sentence of the abstract: “Animals are able to rapidly infer, from limited experience, when sets of state-action pairs have equivalent reward and transition dynamics.”? Is it necessary to write here?
2. For me, a straightforward approach to learn homomorphism in discrete action space, is to parameterize value function as $Q(\sigma(s), \phi_s(a))$, then train it end-to-end with some Q-learning loss, and also train $\sigma$, $\phi_s$ with MSE losses to satisfy (1)(2). It is similar to the approach in the continuous action version of homomorphism (Rezaei-Shoshtari et al., 2022). The action can be one-hot encoded. I am a bit surprised that this simple approach is not discussed in the paper, and what do you think of this approach?

---

> ### Author Response · Authors · 2023-11-17
> **response to  YpK3**
>
> Thank you for taking the time to review our work. **One major criticism is that we need more experimental verification. We agreed with this criticism. We have now run all the MinAtar games with the model now being learned online with a new model-based baseline also included.**
>
> **Regarding limiting assumptions**
>
> We agree deterministic dynamics is limiting, which is why we also validate our approach in stochastic environments (pred-prey and Asterix). Taking your advice onboard, we have further addressed this limitation by adding more MinAtar games.
>
> For the state dependent rewards, we agree that this is a limitation of our work. However, our approach is still applicable to a range of discrete control environments (e.g. all the MinAtar games). If there is an important class of environments we have not listed in our limitations, please let us know so that we can include it.
>
> **Regarding your proposal for learning a homomorphism by directly optimizing on the definition of an MDP homomorphism with an MSE loss**
>
> Thank you for this suggestion but, as far as we are aware, this would not work. Only optimizing the MSE loss based on the MDP homomorphism would mean there is nothing to prevent the homomorphic state mapping from learning a trivial representation (e.g. mapping all states to a vector full of zeros would give zero loss). Van der Pol et al. (2020a) back up this claim explicitly (see page 3 left column of their paper) and get around this issue by using a contrastive loss (which we compare to in the Cartpole and Predator Prey experiments, however, there has not been any attempt to scale their dynamic programming approach to larger inputs, which is why we do not compare to it in MinAtar).  Rezaei-Shoshtari et al. (2022) get around this by adding an approximate bisimulation metric to their loss function in addition to the MDP homomorphism loss function. We think it could be possible to balance these two terms in the discrete setting but would be a significant project in its own right.
>
> **Regarding clarity and mistakes in the writing**
>
> - return is now in expectation
> - $\sigma$ is now always a function of $(s, a)$, as our mapping is action dependent
> - removed the hat in the algorithm
> - figure 3(a) typo is fixed
> - starting sentence with \forall has been fixed
> - double which on page 3 fixed
> - we have fixed the issues with \citep and \citet
>
> all of these changes are in the updated manuscript.

---

> > ### Comment · Reviewer_YpK3 · 2023-11-20
> >
> > Thanks for your response that resolves some of my concerns! But still some core concerns remains:
> >
> > - Define the symbols like \sigma and \alpha_s explicitly with inputs and outputs. Also provide some textual description. When you define $h$, there is a typo of >}
> > - I think MDP homomorphisms is defined with a $\sigma(s)$ on state representation, but not $\sigma(s,a)$ used in this paper. See Van der Pol et al. (2020b). Your paper also states that "Where σ is a mapping from experienced states to abstract states". I doubt if your current formulation follows the classic definition
> > - It is still hard for me to understand the hypothetical action part. For example, in Eq. 6, LHS is state representation while RHS generated by the backward model is a state, why these two can be equal?
> > - About the new experiments: I appreciate the authors running more complicated tasks (which I requested before). The results show that EEA is much better than DQN in 0.5e6 sample regime, but worse in larger sample regime. Although this is not surprising as the tasks breaks the EEA's assumption, it is bit discouraging. I think a more favorable benchmark (composed of many tasks, more high-dim than cartpole) should follow your method's assumption to show the strength.
> >
> > Therefore, I maintained my rating.

---

> ### Author Response · Authors · 2023-11-22
>
> Thanks for giving us more feedback to improve our paper. We really appreciate it. We have adressed your points as follows (let us know if it improves things).
>
> - we provide textual description and explicit input, output function signatures for $\sigma$ and $\alpha$
> - our paper is now consistent in that all $\sigma$ are written as $\sigma_{a}$ to note the dependent on actions, this is different from the Van der Pol definition and we make note of it. Is that satisfactory?
> - Apologies for not being clear on this, now in our homomorphism definition we describe how our abstract states have the same dimensionality as they are predicted as states that you would see from your environment. This means the LHS and RHS can be equal in Eq. 6.
> - could you suggest a benchmark?

---

### Official Review · Reviewer_QLTL · 2023-10-31

**Soundness:** 1 poor
**Presentation:** 2 fair
**Contribution:** 2 fair
**Rating:** 3
**Confidence:** 4

**Summary:**

This paper proposes an interesting method for learning MDP homomorphisms to reduce sample complexity. The proposed method uses a forward model and a backward model to abstract state action pairs, with hypothetical actions. Some experiments on tabular environments and coutinuous environments are done to verify the effectiveness of the learned homomorphisms.

**Strengths:**

The idea of learning homomorphisms with forward and backward models is interesting, although the concepts are not new.
The method has the potential to scale up in large visual RL problems, where state abstraction is important.

**Weaknesses:**

1. There are several important points not clarified or explained in the paper. For example, how can we learn good forward and backward models with function approximators? In theory, accurate forward and backward prediction requires complete coverage of the state action space, which can be even more expensive than learning a policy (without abstraction). But the algorithm directly assumes the availability of such models, which is unrealistic.
2. The success of the abstract largely depends on the correctness of the forward and backward models. If any of them give incorrect or biased prediction, the learned homomorphisms and values will fail. Combined with point 1 above, I am not convinced how the proposed algorithm can be widely applicable.
3. Although the paper tries to demonstrate empirical results on different benchmarks, the experiment still looks a bit limited. Only Q learning or DQN are considered as the base algorithm. Can it be combined with other RL algorithms such as PPO or SAC?
4. The return on Asterix is much lower than reported by prior papers. The original nature DQN paper has return 6012 on Asterix, but why is the return reported by this paper below 20? Is there any rescaling of the reward?
5. From experimental results, the improvement made by the algorithm is mainly on the converging speed or sample complexity. However, I am not sure whether it is a fair comparison with baselines, since the proposed method trains the policy over a pretrained forward/backward model, which already consumes a lot of samples.
6. The notations are not clear and sometimes confusing. For example, the hypothetical actions are sometimes referred to as $a_{hyp}$ and sometimes as $\hat{a}$. In Eq (7), $a^\prime$ is never defined. $\sigma$ is sometimes a function of states, sometimes a function of state-action pairs. In Eq (8), should the first $s^\prime$ be $\bar{s^\prime}$? Although the meanings can be roughly inferred from the context, it is still not reading-friendly.

**Questions:**

- Is there any experimental result with learned forward/backward model on large-scale environments?
- Can the forward/backward model be learned together with the policy?

---

> ### Author Response · Authors · 2023-11-17
> **response to QLTL**
>
> We sincerely appreciate the time you have taken to review our paper and give us some feedback to improve on.
>
> **Regarding more MinAtar results with learned models**
>
> The criticism that sticks out to us most seems to be that we do not have sufficient results where the transition models are learned online. As a result, we have rewritten the MinAtar sections with new results where the transition model is learned online for all MinAtar games for a fair comparison with the baselines.
>
> We found different optimizers work best for different algorithms and environments. To take this effect into account, we compare the ratio of performance improvement from EEA to a standard DQN for each optimizer and average over optimizers (RMS and Adam). We also average the performance improvement over all games. This is the result shown in Figure 4 in the updated manuscript. As you can see, overall, EEA delivers a nice performance boost in the low sample limit but its asymptotic performance is slightly worse after a large number of frames.
>
> **Regarding Asterix Returns**
>
> It is true that the returns are different to the normal Atari Asterix returns. This is because we are using MinAtar and there is not a one to one mapping between the returns in Atari (Bellemare et al. 2013) and MinAtar (Young and Tian 2019).
>
> **Regarding comparisons to SAC and PPO**
>
> It is true that we only compare to value based methods. A limitation of our approach is that it is currently only formulated for actor-critic methods on discrete control tasks. We think this is not so bad as the state-of-the-art for sample efficient discrete control is value-based (Schwarzer et al. 2023).
>
> **Regarding lack of clarity in notation**
>
> We thank you for pointing out some lack of clarity in our notation. We have addressed it in the following ways:
>
> - $\sigma$ is now always defined as a function of states and actions.
> - we now always refer the hypothetical action as $a_{hyp}$
> - we define a’ in our definition of the MDP homomorphism
> - we clarify in the text that in equation 8, $s’=\bar{s}’$ because both transitions lead to the same next state
>
> All of these changes are in the updated manuscript.

---

> ### Comment · Reviewer_QLTL · 2023-11-21
> **Reply**
>
> Thank you for the response and the updated results. But my concerns are not fully addressed. The contribution is still limited for ICLR both in terms of theoretical insights and empirical results. So I would maintain my current rating.

---

> ### Author Response · Authors · 2023-11-22
>
> Thank you for engaging with us!
>
> As we understood it, the lack of learned models was one of the main issues you found with the paper (3/6 of the main weaknesses were related to this). We also think the Asterix return issue was cleared up. Finally as suggested,  we updated the notation to be more reader friendly regarding hypothetical actions.
>
> As we see it that just leaves the PPO/SAC comparison weakness from your main review. We acknowledge this is a weakness we haven't addressed.
>
> Nevertheless, there is a mismatch in our understanding of your criticism, we feel 5/6 of the weaknesses have been adressed while you say your concerns still remain. If possible, could you mention which of the concerns still remain and how they could be adressed?

---

### Official Review · Reviewer_NiFy · 2023-11-01

**Soundness:** 1 poor
**Presentation:** 1 poor
**Contribution:** 1 poor
**Rating:** 3
**Confidence:** 4

**Summary:**

This paper is concerned with sample efficiency in reinforcement learning (RL). The paper argues that RL systems can achieve greater sample efficiency by using a type of homomorphism they call Equivalent Effect Abstractions (EEAs).

Presumably, EEAs permit one to map between representations of environment state while preserving the original reward structure. Provided that such mappings exist, can be efficiently implemented, and one can map multiple environment states to a single encoding, then a learning system can plausibly use the encodings to generalize policy evaluations / improvements over states.

The paper makes several claims.
1. It is possible to efficiently learn EEAs.
2. Learning EEAs requires no prior knowledge of reward structure.
3. EEAs reduce the size of the state-action space "by a factor equal to the cardinality of the action space."
4. EEAs can be used to improve the sample efficiency of policy learning.

I am recommending this paper is rejected, as it lacks support for all of its claims. Specifically, the paper contains little to no support for 1, 2, and 3. And the empirical support for claim 4 is not convincing. Moreover, I believe these issues are too substantial to be addressed with a simple revision.

**Strengths:**

* Sample efficiency is indeed an area that can benefit from more research.
* Exploiting symmetries and equivalences in the reward structure seems like a promising approach to designing new sample-efficient algorithms.
* Methods applicable to deep reinforcement learning stand to have high significance addressing large-scale problems.

**Weaknesses:**

(Technically unsound) The paper does not contain enough information for a reader to fully understand what is being proposed.
* Formalism lacks the precision needed to understand the main technical concepts. For instance, a "forwards model" and "backwards model" are never defined. These seem like parameterized functions. However, their input-output signatures are not described, and their parameters seem to change from Equation 6 to Algorithm 1.
* The paper fails to describe how EEAs, and its associated learning process, fit within the standard RL framework of action, observation, and reward. This is essential for others in the community to use the idea---to know how a step of policy evaluation or improvement is performed.
* The scope with which EEAs are presented is too broad, and this gives EEAs a somewhat nebulous identity. According to the paper, they apply to both tabular and deep settings, and settings where models are both available and learned. However, the paper provides no specific description of how the idea was instantiated in each respective setting. Ironically, the little information that is used to define EEAs limits their scope to a restrictive degree.
* The paper fails to establish EEAs as a distinct idea, separate from prior work in homomorphic MDPs (van der Pol et al., 2020) and using bisimulation methods.

(Technically unsound) The empirical results do not provide sufficient support for claim 4.
* Current methodology fails to account for the pre-training experience used to learn the EEA.
* Experiments are missing important baselines. For instance, a standard model-based learner that doesn't also learn a homomorphism.
* Experiments use an inconsistent methodology between baselines---for example, different numbers of seeds.
* Experiments show learning curves, but should go further---to translate them into metrics of sample efficiency, such as steps of experience (accounting for pre-training).

(Limited significance) The main contribution is potentially limited, as it only applies to fully-observable, deterministic MDPs where rewards are strict functions of state (i.e. no action dependence.)

(Limited significance) Related work does not make connections to paper's main ideas.

(Poor presentation) Mathematical notation is sometimes overloaded and often undefined.

**Questions:**

* Many references are missing parentheses. This makes some sentences difficult to read.
* The current manuscript points readers to external references to learn about homomorphisms. You could broaden your audience and add clarity to the paper by introducing the concept in the introduction.
* A better reference to use for introducing the MDP formalism is Sutton and Barto (The RL book), or Putterman's MDP book.
* Seem to be based on the assumption that different states that transition to the same state have the same values.
* This is not generally true:
  > Equivalent effect abstraction is based on a simple observation that is guaranteed to be true in the majority of RL problems if R and P are deterministic.

---

> ### Author Response · Authors · 2023-11-17
> **response to NiFy**
>
> Thank you for taking the time to review our paper in detail. Based on your feedback, we have made some changes, which we believe improves the quality of the paper.
>
> **Regarding technical unsound comments**
>
> - We have run our approach on all MinAtar games where we learn the model online (and so account for model learning experience)
> - We add a model-based Dyna baseline to compare to for MinAtar
> - We put our results into table describing results after a given number of steps
>
> **Regarding limited significance**
>
> While it is true our approach is defined theoretically for deterministic MDPs, we test what happens when we break this assumption by running on stochastic environments on predator prey and the MinAtar environments.
>
> **Regarding clarity and mistakes in writing**
>
> - We now provide more details on the input and output signatures of our models after equation 5.
> - We now describe explicitly that all models are trained with mean squared error loss and experience tuples from the DQN experience replay buffer
> - After we now describe explicitly how you obtain a policy from our approach
> - We have addressed the issues with parentheses in our citations
> - One claim is that we have not differentiated ourselves from Van der Pol (2020a or b?), we think our approach is quite different from both and we describe their approach when we introduce the baselines. Can you provide more detail about why they are not seen as distinct from our approach in the paper?
> the parameters change from algorithm 1 to equation 6 is a typo we have fixed
>
> Lastly, if possible, could you provide some more detail or example benchmark environments where our assumptions about value equivalence are not true? That way we can understand your criticism on this point better.

---

> > ### Author Response · Authors · 2023-11-22
> >
> > Have you had a chance to read our rebuttal and updated PDF?
> >
> > We are keen to know if it adresses any of your criticisms.

---

### Official Review · Reviewer_1RGv · 2023-11-01

**Soundness:** 3 good
**Presentation:** 3 good
**Contribution:** 1 poor
**Rating:** 5
**Confidence:** 3

**Summary:**

MDP homomorphisms can be used to increase sample efficiency in reinforcement learning. Prior work has often used domain knowledge of symmetries to build these homomorphisms. The authors present a method for constructing them from experience, called equivalent effect abstraction, which uses the idea that actions that lead to the same next state often have the same value. The authors validate equivalent effect abstraction by testing its effect on the efficiency of value-based learning in a variety of RL environments: Tabular RL Maze, Cartpole, Stochastic Predator Prey, and MinAtar Asterix.

**Strengths:**

* The paper is written clearly, and the problem is well-motivated. Leveraging a backwards model to find actions with similar effects is clever and insightful.
* Equivalent effect abstraction is evaluated in several different types of environments, and the experiments conducted seem sound.
* In games well-suited to equivalent effect abstractions and where hypothetical actions are easy to define, empirical results suggest a positive effect on sample efficiency.
* Despite my comment in the *Weaknesses* section, the authors do a good job discussing the potential limitations of the approach related to stochastic state transitions and some of the complications that arise from edge cases when selecting a hypothetical action.

**Weaknesses:**

My main concern is the applicability of this approach beyond the domains that have been tested to showcase it. The approach depends on selecting a hypothetical action and the authors mention that one must be selected carefully in certain domains, but do not elaborate sufficiently. The domains selected for experiments seem well-suited to equivalent effect abstraction because they are generally spatial "navigation" tasks in 2D - I'm not sure how to describe the domains precisely, but where hypothetical actions are easy to select and equivalent effects are prominent. It would be interesting to see this idea applied in the other MinAtar games. The preliminary results in the supplementary material seem to suggest applying this approach may not be as straightforward in Breakout, for example.

**Questions:**

Why was asterix chosen as the only benchmark tested from MinAtar?

---

> ### Author Response · Authors · 2023-11-17
>
> Thank you for your review and appreciate that you found our work clever and insightful. We address the weaknesses you found in our work below and hope having addressed your concerns we have improved the paper.
>
> **“My main concern is the applicability of this approach beyond the domains that have been tested to showcase it … I'm not sure how to describe the domains precisely, but where hypothetical actions are easy to select and equivalent effects are prominent. It would be interesting to see this idea applied in the other MinAtar games.”**
>
> We agree that this is a useful next step. As a result, we have rewritten the MinAtar section of our paper with new results on all games with a model that is learned online. We find that the benefit is most obvious in games like Asterix and Seaquest, where the number of actions collapsed down to the hypothetical action is large. Nevertheless, we also see a nice improvement on games like freeway when using the RMS optimizer, where the hypothetical action is not easy to select.
>
> **Why was asterix chosen as the only benchmark tested from MinAtar?**
>
> Asterix was initially chosen as a benchmark because it has a large number of equivalent actions, so hence is a good demonstration of our approach. Since our initial submission we have run EEA on all of the Minatar games. We hope this addresses some of your concerns about the general applicability of our approach. We applied EEA to some games where the action structure isn’t as simple (e.g Seaquest which has a “fire” action or Freeway where the speed of moving forward depends on previous actions). While some of these results are positive, we do agree that our approach is very well suited to navigation tasks.

---

> > ### Comment · Reviewer_1RGv · 2023-11-22
> >
> > I acknowledge I've read the other reviews and the author responses. The other reviewers raise justifiable doubts about the paper's contribution (both theoretical and empirical) and clarity, which the authors do not sufficiently address. I've updated my score with that in mind.

---

> > > ### Author Response · Authors · 2023-11-22
> > >
> > > Thanks for engaging, we were disappointed to see you lowered your score after we added experiments you suggested.
> > >
> > > Is there a particular criticism from the others that is troubling for you?

---

### Author Response · Authors · 2023-11-18
**New MinAtar results on all games with a learned model**

In response to criticism from the reviewers, we have performed more experiments on the MinAtar suite.

- we now train the forwards and backwards models continously alongside the Q-network for the MinAtar experiments
- we now test our approach on all MinAtar games
- we have a new dyna baseline for the MinAtar results that we compare to

We think this improves the paper and request that the reviewers read the new revised MinAtar section in addition to our reviewer specific rebuttals. Overall, when averaging over different optimizers and games, we find our approach gives an almost 4x improvement in normalised scores in the low sample (250k) limit.

We have also addressed specific concerns from reviewers in the updated manuscript and provide details on this in the individual rebuttals.

---

### Meta-Review · Area_Chair_CEHn · 2023-12-06

**Metareview:**

This paper presents a mechanism to leverage MDP homomorphisms to increase sample efficiency in RL. While the paper explores an interesting and important problem, the proposed method makes very restrictive assumptions about the MDP (fully-observable, deterministic MDPs where rewards are strict functions of state). The reviewers also raised concerns about the clarity of the exposition and the applicability of the empirical results. We appreciate the changes made by the authors in response to the reviews, and encourage them to examine if weakening the assumptions is possible, or if more interesting experimental domains can be found where the assumptions still hold.

**Justification For Why Not Higher Score:**

Overall all the reviewers seem to agree that the experimental results do not sufficiently support the main claims of the paper. The core assumptions seem too restrictive to make the proposed method widely useful.

**Justification For Why Not Lower Score:**

N/A

---

### Decision · Program_Chairs · 2024-01-16

Reject